# Adversarial for Good? How the Adversarial ML Community's Values Impede Socially Beneficial Uses of Attacks

**Kendra Albert** [* 1]  **Maggie Delano** [* 2]  **Bogdan Kulynych** [* 3]  **Ram Shankar Siva Kumar** [* 4]

## Abstract

Attacks from adversarial machine learning (ML) have the potential to be used "for good": they can be used to run counter to the existing power structures within ML, creating breathing space for those who would otherwise be the targets of surveillance and control. But most research on adversarial ML has not engaged in developing tools for resistance against ML systems. Why? In this paper, we review the broader impact statements that adversarial ML researchers wrote as part of their NeurIPS 2020 papers and assess the assumptions that authors have about the goals of their work. We also collect information about how authors view their work's impact more generally. We find that most adversarial ML researchers at NeurIPS hold two fundamental assumptions that will make it difficult for them to consider socially beneficial uses of attacks: (1) it is desirable to make systems robust, independent of context, and (2) attackers of systems are normatively bad and defenders of systems are normatively good. That is, despite their expressed and supposed neutrality, most adversarial ML researchers believe that the goal of their work is to secure systems, making it difficult to conceptualize and build tools for disrupting the status quo.

## 1. Introduction

Adversarial machine learning (ML) is a subfield of ML that studies *adversarially*-mounted *attacks* against ML-based systems. The most common research subject in the field is improving *adversarial robustness*: studying and finding

---
[*]Equal contribution  [1]Harvard Law School, Cambridge, MA, USA [2]Engineering Department, Swarthmore College, Swarthmore, PA, USA [3]EPFL, Switzerland [4]Microsoft, Redmond, WA, USA. Correspondence to: Ram Shankar Siva Kumar <ramk@microsoft.com>.

*Accepted by the ICML 2021 workshop on A Blessing in Disguise: The Prospects and Perils of Adversarial Machine Learning.* Copyright 2021 by the author(s).

ways to protect ML models against these attacks and adversaries. If we take a step back, however, we can see that protecting ML is not a goal that is universally desirable and agreed-upon. Ubiquitous applications of predictive algorithms such as ML, e.g., in credit monitoring (Citron & Pasquale, 2014), facial recognition (Buolamwini & Gebru, 2018), and hiring (Sánchez-Monedero et al., 2019) regularly cause harm and offence in the real world, and consequentially meet resistance in various forms: policy decisions and legal frameworks (European Commission, 2020), activism (Belfield, 2020), and even artistic interventions (Doringer & Felderer, 2018).

Besides these forms, harmful technological systems can also be resisted by leveraging technological tools in a subversive way. For instance, within the security and privacy community, there have been numerous technological proposals for protecting against privacy-invasive ML systems and their supporting infrastructures (Brunton & Nissenbaum, 2015), including systems for protecting people's privacy against facial recognition (Shan et al., 2020; Chandrasekaran et al., 2020), voice recognition (Chandrasekaran et al., 2019), and mechanisms for releasing information while preventing inferences about private attributes (Jia & Gong, 2020; 2019). Outside of privacy protection, there are proposed systems for externally rectifying unfair outcomes, and for influencing harmful ML systems from the outside in the absence of other leverage (Kulynych et al., 2020; Vincent et al., 2021; Delobelle et al., 2021).

Most of the proposals above make use of techniques from adversarial ML such as adversarial examples and poisoning attacks. In fact, adversarial ML offers a natural toolbox for protesting, contesting, and reconfiguring outcomes of ML systems from the outside (Kulynych et al., 2020; Albert et al., 2020b; Das, 2020). Why despite this significant body of work is it so rare that mainstream adversarial ML researchers see the ML model itself as an adversary? Why is it that research on these non-trivial applications of adversarial ML comes almost exclusively from outside of the core adversarial ML community? We argue that the fundamental values of the adversarial ML community has limited its vision with regards to the beneficial potential of these technologies.

In philosophy and sociology of science, it is widely agreed that culture, experiences, and — crucially — *values* of an academic community have a significant impact on which research problems are considered, and which ones are not (Latour, 1993; Haraway, 1988; Kuhn, 1970). Clearly, some of these values, called *epistemic* (Carrier, 2013), are related to knowledge and its production, e.g., a community can value accuracy, mathematical elegance, or novelty. Epistemic values, however, are not the only ones that shape research. Even though the scientific process is commonly seen as a neutral "gaze from nowhere" (Haraway, 1988), other values, called *non-epistemic*, also have an impact. The non-epistemic values are often not explicitly stated, but include underlying social, moral, or political values, e.g., a community can value applicability to human needs (Longino, 1995).

Can we assess the values of the adversarial ML community? The fact that the community does not engage with a variety of high-profile targets to build attacks "for good" must be of moral, social, and political nature, thus due to non-epistemic values. Luckily for us, the NeurIPS conference, following existing practices by e.g. the US National Science Foundation, has introduced the requirement for inclusion of broader impact statements in all submissions in 2020:

*"In order to provide a balanced perspective, authors are required to include a statement of the potential broader impact of their work, including its ethical aspects and future societal consequences. Authors should take care to discuss both positive and negative outcomes."* (NeurIPS, 2020a)

The stated ethical and societal impacts of a given work are likely to reflect some of the moral and social values held by the authors. This presents us with an opportunity to get a glimpse of the community's values through the lens of the broader impact (BI) statements. In this paper, we delve into the BI statements in adversarial ML papers at NeurIPS 2020 to see how the authors view the impacts of their work, gain understanding of what values the community holds, and whether these values are compatible with the direction and the promise of "Adversarial for Good."

## 2. Methodology

In this section, we outline our methodology for selecting papers on adversarial ML at the NeurIPS 2020 conference, and our content analysis protocol.

**Relevance**. The object of our study were NeurIPS 2020 papers that discuss security-relevant adversarial settings in which the adversary's goal is to interfere with the functionality of ML models. The in-scope topics are adversarial examples, poisoning attacks, model stealing, as well as defences against these. The out-of-scope topics were the "adversarial" optimization techniques involving multiple competing objectives, as in generative adversarial networks (GANs),

and works on distributional robustness that do not evaluate or discuss adversarial robustness. Additionally, we deemed attacks against privacy of the training data such as membership inference (Shokri et al., 2017) as out of scope.

**Selection**. To obtain information about the papers, we retrieved the data from the NeurIPS 2020 website containing details about all 1918 accepted papers (NeurIPS, 2020b). We used a two-stage process to select the relevant ones. First, we automatically filtered the papers by occurrence of the following non-case-sensitive character sequences in their titles: "adversar" (to capture "adversary" or "adversarial"), "robust" (to capture "robust" and "robustness"), "poison", "steal", "attack"; and excluded occurrences of the sequence "generative adversarial." This stage yielded 154 papers. We further manually inspected each of the initially filtered papers to ensure they fit our relevance criterion. Our final tally resulted in a total of 91 papers with a breakdown of 16 *spotlight* papers (a special distinction at NeurIPS, reflecting high scores at the review stage) and 75 non-spotlight papers.

**Coding**. Having selected the relevant subset of papers, our goal was to better understand what the authors consider the impact of their work to be, and shed the light on their values. For this, we followed a standard protocol for qualitative data analysis known as inductive coding (Berelson, 1952): a set of *coders* manually assigned *codes*, i.e., labels of interest, to all the papers. Here, the set of coders is all the authors of the present manuscript. To come up with the codes initially, all four authors read most of the spotlight papers and took notes on themes. After this initial review, we determined that there were three assumptions that seemed to cut across the spotlights, two of which we used as the main codes.

The first assumption is what we call "robustness as a final value." The underlying normative value is that ML systems *should be* adversarially robust — as in, resistant to attacks. BI statements consistent with this value often contain the assumption that adversarial robustness is the end goal of adversarial ML research, often noting that robustness is a prerequisite to AI-systems being employed in "high risk" or "safety-critical" applications.

The second assumption we noted as common across the field is the moral judgement that attackers of systems are bad, and that defenders of systems are good. This corresponds to robustness as a normative good, but is different insofar as a paper might acknowledge that not all attacks are bad but still believe that adversarial robustness is net good.

The third assumption we observed was the idea that technology itself is neutral, and therefore does not have political properties nor relevant considerations for BI statements. We ended up not using this assumption for coding for reasons described shortly.

We also included questions about whether or not a BI statement discusses negative impacts, limitations of their work, ethical issues with their research, and/or mentions possible downstream applications. Moreover, we added questions about the form of the BI statement: whether the statement is only restating conclusions/future work, and whether it cites other sources. Finally, we also gathered information about the funding sources.

After the preliminary review of the spotlight papers, we proceeded with a two-stage coding process as presented in Davis et al. (2014). First, all coders went back and independently coded the spotlight papers that were initially used to create the set of values evaluated in the paper. This process helped us refine the questions and codes. For instance, one of our initial questions was to quantitatively assess how much of the BI statement was treated as conclusions or future work. We changed this from a percentage to a Yes/No question asking if the BI statement was *primarily* used to discuss conclusions and future work. We also dropped the question "Does the BI statement assume technology is neutral?" from our analysis as it was difficult to reliably assess from reading the BI statement. In the second stage, each of the non-spotlight papers were randomly assigned to two coders. This way, each paper was reviewed at least twice.

Once the papers were all coded via the form, we examined those papers that lacked consensus among the reviewers. For spotlight papers, which were reviewed by all four authors, if two or more coders disagreed even in one of the responses, we deemed it a "conflict"; for the non-spotlight papers, which were reviewed by two coders, there was a conflict if either of the coders disagreed in their responses. We resolved conflicts by discussing the paper with other coders, debating the answers we initially chose until we came to consensus.

## 3. Findings

Of the 91 papers reviewed, 88 papers had BI statements. The 3 papers without BI statements included a comment that their paper was theoretical and/or did not have any foreseeable societal consequences. However, even on papers without a broader impacts statement, meta-reviewers discussed the positive impacts that these works might have on the field, which contradicts the idea that these works are theoretical and have no societal consequences unless one assumes that the field is divorced from society.

**Perceived Broader Impact**. The 88 papers that included a BI statement ranged in their level of engagement with broader impacts. In particular, only one-third of the papers (30/88) went beyond restating future work and conclusions from the paper itself. About half of the papers (45/88) discussed negative societal impacts, the most common of

*Table 1.* Summary of findings from the qualitative coding NeurIPS 2020 Adversarial ML papers. The full text of the questions is included in the Appendix.

| Question | Yes | No | N/A | Total |
|---|---|---|---|---|
| Paper has BI statement? | 88 | 3 | 0 | 91 |
| BI statement primar. restates conclusions? | 58 | 30 | 0 | 88 |
| Security as neg impact in BI? | 23 | 65 | 0 | 88 |
| Societal harms as neg impact in BI? | 13 | 75 | 0 | 88 |
| Environ. harms as neg impact in BI? | 5 | 83 | 0 | 88 |
| Broad harms of field as neg impact in BI? | 13 | 75 | 0 | 88 |
| No neg. impact in BI | 43 | 45 | 0 | 88 |
| Limitations of paper in BI? | 22 | 66 | 0 | 88 |
| Applications of their work in BI? | 36 | 52 | 0 | 88 |
| Cite other work in BI? | 28 | 60 | 0 | 88 |
| Ethics of conducting their research in BI? | 5 | 83 | 0 | 88 |
| "Robustness as Final Value" assump. in BI? | 62 | 11 | 15 | 88 |
| "Attack Bad, Defend Good" assump. in BI? | 38 | 8 | 42 | 88 |
| Paper funded by Military? | 32 | 59 | 0 | 91 |
| Paper funded by Govt. (non-Mil.)? | 60 | 31 | 0 | 91 |
| Paper funded by Industry? | 28 | 63 | 0 | 91 |
| Paper funded by Academic Fellowship? | 7 | 84 | 0 | 91 |
| Paper funded by Non-profit/Other? | 9 | 82 | 0 | 91 |
| Paper received no external funding? | 2 | 89 | 0 | 91 |
| Paper funding not specified? | 8 | 83 | 0 | 91 |

which was security concerns (23 papers). The most common security concern was the potential for published attacks to be used by "adversaries" in ways that would result in negative social outcomes. A total of 13 papers cited non-security related societal impacts in particular, and 13 papers referenced the impacts of the "broader field" (for example, by referencing the field of computer vision broadly). Five papers discussed environmental impacts due to computation. A total of 22 papers discuss potential limitations of their present work in the BI statement. Less than half the papers (36/88) discuss a specific example of how their work might be used in the future. Those that did primarily discussed the potential of adversarial robustness to make ML safe for autonomous vehicles and healthcare settings. About a quarter of the total papers cited other work (28/88 papers). This suggests that authors may not be citing the papers that influence their thinking, or may not be engaging with literature as part of writing BI statements.

Surprisingly, given the broader context of BI statements being part of an evaluation of ethical practices within the field, only five papers out of 88 papers explicitly considered the ethics of conducting the research itself. Most adversarial machine learning papers require extensive computing resources (Luccioni et al., 2020), and some may include human subjects testing or be done in other settings where there are direct ethical concerns about the research (Albert et al., 2020a).

**Values**. In the BI statements that we reviewed, most authors do not consider (or at least do not mention) that robustness may not be normatively desirable if an ML system is being

used for anti-democratic or unethical purposes, nor do they weigh the pluses and minuses of limiting the possibility of circumvention of such systems. Over 70% of papers (62/88) had robustness as a final value. Only 11/88 of the papers did not have robustness as a final value, with the remaining 15 papers listed as not applicable as it could not be assessed one way or another. The papers that did not hold robustness as a final value often noted that whether robustness was normatively desirable depended on the context in which the system was employed. For example:

*"This uncertainty is symptomatic of the fact that machine learning is often fundamental by nature and that there is no machine learning technique for improving robustness that can be applied only to positive-impact applications, whatever one's subjective interpretation of 'positive' may be."* (Yang et al., 2020).

The assumption that attackers were "bad" was borne out in a number of ways: for example, the use of language like "malicious" or "malevolent" was often used to describe users of attacks, or attacks were described as aligned with "negative goals" (Elinas et al., 2020). A little less than half of coded papers explicitly cited attackers as bad and defenders as good (38/88), but many did not comment one way or the other, for a total of 42 not applicable. Only 8 of papers that expressed an opinion rejected the idea that attackers are bad and defenders are good.

Although we did not end up using our study results from the question as to whether technology is neutral, it is worth mentioning that many papers adopted an agnostic view as to the pluses and minuses of the technologies produced. The Yang et al. quote above, for example, while rejecting robustness as a normative good, makes this neutrality assumption. Although machine learning may be fundamental, that does not mean that it is "neutral": as any technology, it can have a tendency to benefit some actors at the expense of others (Winner, 1980). For instance, speeding up the processing of large datasets and making predictions from them fundamentally benefits those with access to more data (Dotan & Milli, 2019; Albert et al., 2020b).

**Funding**. Two thirds of the papers received government funding (60/91), a little over one third of the papers received military funding (32/91), and about one third of the papers were funded by industry (28/91) (papers can and often were funded by more than one source). Additional funding sources included academic fellowships (7 papers) and non-profits (9 papers). Eight papers had no funding specified. Two papers explicitly listing that no third party funding was obtained. These results suggest that government and military funding drive many significant advances in this field. Although it would be overly simplistic to say, for example, that military funding directly results in technologies that can be applied offensively, funding of research can drive values

and priorities, and researchers generally did not discuss how the funding of their work might affect its broader impacts.

## 4. Conclusions

If broader impact statements are an accurate statement of values, most adversarial ML researchers publishing at NeurIPS 2020 believe that building more robust, reliable machine learning systems is the goal of the field, and that increased security of ML systems is a positive impact, independently of context. This is concerning because progress in ML is heavily laden with socio-political and environmental values. These values are manifested in and caused by the reliance and strong compatibility of ML theory and methods with "compute-rich and data-rich environments" (Dotan & Milli, 2019). Albert et al. (2020b) and Alkhatib (2021) argue that one of the declared purposes of ML systems is to leverage data collected about a deployment environment, often by outsiders, to better understand that environment. Making environments "legible", or more easily understood, means that those environments are more susceptible to influence by centralized power structures such as governments, who often have disproportionate access to the data and computational resources needed for deployment of ML systems. This is concerning not only because it can increase the power of state actors, but also because these models can give a false sense of confidence in an understanding of the environment in which they are deployed (Scott, 1998). Our findings suggest that adversarial ML shares these underlying values with the core field of ML, rather than being adverse to it. This creates a significant challenge for efforts to use adversarial ML attacks "for good", because the hidden values of the field may keep researchers from even realizing that the production of useful subversive technologies is something that adversarial ML is uniquely well-suited to tackle.

Although we primarily discussed attacks in adversarial ML as a means to disrupt the operation of harmful systems or re-configuring their outcomes, these are not the only beneficial applications of attacks. For example, Albert et al. (2020b) argue that techniques from adversarial ML can provide us with tools for testing for model bias, inclusion of training data without consent, and predatory inclusion, i.e., creation and use of algorithms that nominally support marginalized groups but actually exploit them (Taylor, 2019). In order to produce efficient tools aligned with any of these goals, however, researchers in adversarial ML need to fundamentally reframe the goals of the field. This requires understanding machine learning as neither a neutral technology nor a straight-forwardly positive one, but rather one that is uniquely compatible with centralized power, and as such, will require concrete efforts to resist.

## Acknowledgements

The authors would like to thank Maksym Andriushchenko for helpful feedback on the initial version of this paper. They would also like to thank Apryl Williams for assistance with the methodology.

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

## A. Appendix

For the full list of 91 papers that were coded, see the next page.

Papers were manually coded by the authors using Google Forms. We asked the following questions:

- Does the paper have a BI statement?

- Does the BI statement focus primarily on future work or restate conclusions?

- Does the BI statement discuss any negative societal impacts?

- Does the BI statement discuss limitations of the present research?

- Does the BI statement provide specific examples of uses?

- Does the BI statement cite out to other work?

- Did the BI statement discuss any ethical concerns about conducting the research?

- Does the BI statement have "robustness as a final value" as an assumption?

- Does the BI statement have "attackers are bad and defenders are good" as an assumption?

- Does the BI statement assume technology is neutral?

- Funding source for the paper

*Table 2.* List of all the NeurIPS 2020 papers included in the coding analysis.

| NeurIPS Paper ID | Paper Title | Authors |
|---|---|---|
| 17743 | Most ReLU Networks Suffer from $\ell^2$ Advers... | Amit Daniely et al. |
| 18384 | Certified Robustness of Graph Convolution Netw... | Hongwei Jin et al. |
| 17118 | Measuring Robustness to Natural Distribution S... | Rohan Taori et al. |
| 17260 | Simultaneously Learning Stochastic and Adversa... | Tiancheng Jin et al. |
| 17984 | Large-Scale Adversarial Training for Vision-an... | Zhe Gan et al. |
| 18876 | Guided Adversarial Attack for Evaluating and E... | Gaurang Sriramanan et al. |
| 18985 | A Single Recipe for Online Submodular Maximiza... | Omid Sadeghi et al. |
| 18377 | Variational Inference for Graph Convolutional ... | Pantelis Elinas et al. |
| 17327 | Simulating a Primary Visual Cortex at the Fron... | Joel Dapello et al. |
| 17264 | Adversarially Robust Streaming Algorithms via ... | Avinatan Hasidim et al. |
| 18492 | Do Adversarially Robust ImageNet Models Transf... | Hadi Salman et al. |
| 19053 | Robust Sub-Gaussian Principal Component Analys... | Arun Jambulapati et al. |
| 16857 | DVERGE: Diversifying Vulnerabilities for Enhan... | Huanrui Yang et al. |
| 17097 | Adversarial Training is a Form of Data-depende... | Kevin Roth et al. |
| 19009 | Beyond Perturbations: Learning Guarantees with... | Shafi Goldwasser et al. |
| 18912 | Robust Deep Reinforcement Learning against Adv... | Huan Zhang et al. |
| 18059 | A Causal View on Robustness of Neural Networks | Cheng Zhang et al. |
| 17941 | Hyperparameter Ensembles for Robustness and Un... | Florian Wenzel et al. |
| 18518 | (De)Randomized Smoothing for Certifiable Defen... | Alexander Levine et al. |
| 17078 | Adversarial Self-Supervised Contrastive Learning | Minseon Kim et al. |
| 18221 | Input-Aware Dynamic Backdoor Attack | Tuan Anh Nguyen et al. |
| 18102 | Adversarial Blocking Bandits | Nicholas Bishop et al. |
| 17781 | Adversarial Distributional Training for Robust... | Yinpeng Dong et al. |
| 18567 | Understanding and Improving Fast Adversarial T... | Maksym Andriushchenko et al. |
| 17027 | Adversarial Bandits with Corruptions: Regret L... | lin yang et al. |
| 18380 | Attack of the Tails: Yes, You Really Can Backd... | Hongyi Wang et al. |
| 17462 | On the Tightness of Semidefinite Relaxations f... | Richard Zhang |
| 17742 | Adversarial Learning for Robust Deep Clustering | Xu Yang et al. |
| 17187 | A Closer Look at Accuracy vs. Robustness | Yao-Yuan Yang et al. |
| 18467 | Adversarial Counterfactual Learning and Evalua... | Da Xu et al. |
| 17905 | Non-Convex SGD Learns Halfspaces with Adversar... | Ilias Diakonikolas et al. |
| 18095 | Optimal Robustness-Consistency Trade-offs for ... | Alexander Wei et al. |
| 17064 | Adversarial Weight Perturbation Helps Robust G... | Dongxian Wu et al. |
| 18244 | Robust Pre-Training by Adversarial Contrastive... | Ziyu Jiang et al. |
| 17517 | Provably Robust Metric Learning | Lu Wang et al. |
| 18758 | Iterative Deep Graph Learning for Graph Neural... | Yu Chen et al. |
| 18237 | Biologically Inspired Mechanisms for Adversari... | Manish Reddy Vuyyuru et al. |
| 19079 | Lipschitz Bounds and Provably Robust Training ... | Vishaal Krishnan et al. |
| 18867 | Automatic Perturbation Analysis for Scalable C... | Kaidi Xu et al. |
| 17515 | Election Coding for Distributed Learning: Prot... | Jy-yong Sohn et al. |
| 16807 | Maximum-Entropy Adversarial Data Augmentation ... | Long Zhao et al. |
| 18266 | Practical No-box Adversarial Attacks against DNNs | Qizhang Li et al. |
| 16872 | Adversarially Robust Few-Shot Learning: A Meta... | Micah Goldblum et al. |
| 17522 | GNNGuard: Defending Graph Neural Networks agai... | Xiang Zhang et al. |
| 16975 | Fast Adversarial Robustness Certification of N... | Sascha Saralajew et al. |
| 17076 | Learning Black-Box Attackers with Transferable... | Jiancheng YANG et al. |
| 17122 | On the Stability and Convergence of Robust Adv... | Kaiqing Zhang et al. |
| 19008 | Adversarial Attacks on Deep Graph Matching | Zijie Zhang et al. |
| 17291 | Contrastive Learning with Adversarial Examples | Chih-Hui Ho et al. |
| 17947 | Adversarial Crowdsourcing Through Robust Rank-... | Qianqian Ma et al. |
| 17567 | Diversity can be Transferred: Output Diversifi... | Yusuke Tashiro et al. |
| 18094 | Robust and Heavy-Tailed Mean Estimation Made S... | Sam Hopkins et al. |

| NeurIPS Paper ID | Paper Title | Authors |
|---|---|---|
| 17791 | On 1/n neural representation and robustness | Josue Nassar et al. |
| 18662 | HYDRA: Pruning Adversarially Robust Neural Net... | Vikash Sehwag et al. |
| 18763 | The Complexity of Adversarially Robust Proper ... | Ilias Diakonikolas et al. |
| 18173 | Adversarial robustness via robust low rank rep... | Pranjal Awasthi et al. |
| 18058 | Targeted Adversarial Perturbations for Monocul... | Alex Wong et al. |
| 18208 | AdvFlow: Inconspicuous Black-box Adversarial A... | Hadi Mohaghegh Dolatabadi et al. |
| 16955 | Once-for-All Adversarial Training: In-Situ Tra... | Haotao Wang et al. |
| 18049 | Adversarial Example Games | Joey Bose et al. |
| 17792 | Boosting Adversarial Training with Hypersphere... | Tianyu Pang et al. |
| 18205 | A Game Theoretic Analysis of Additive Adversar... | Ambar Pal et al. |
| 18005 | On the Loss Landscape of Adversarial Training:... | Chen Liu et al. |
| 18143 | Smoothed Geometry for Robust Attribution | Zifan Wang et al. |
| 18299 | On Adaptive Attacks to Adversarial Example Def... | Florian Tramer et al. |
| 17851 | The Statistical Cost of Robust Kernel Hyperpar... | Raphael Meyer et al. |
| 17285 | Trade-offs and Guarantees of Adversarial Repre... | Han Zhao et al. |
| 17175 | Adversarial Attacks on Linear Contextual Bandits | Evrard Garcelon et al. |
| 17026 | Robust large-margin learning in hyperbolic space | Melanie Weber et al. |
| 18174 | Dual Manifold Adversarial Robustness: Defense ... | Wei-An Lin et al. |
| 18269 | Consistency Regularization for Certified Robus... | Jongheon Jeong et al. |
| 16974 | Backpropagating Linearly Improves Transferabil... | Yiwen Guo et al. |
| 17212 | GreedyFool: Distortion-Aware Sparse Adversaria... | Xiaoyi Dong et al. |
| 17206 | An Efficient Adversarial Attack for Tree Ensem... | Chong Zhang et al. |
| 17181 | Robustness of Bayesian Neural Networks to Grad... | Ginevra Carbone et al. |
| 17871 | Online Robust Regression via SGD on the l1 loss | Scott Pesme et al. |
| 17769 | Adversarial Robustness of Supervised Sparse Co... | Jeremias Sulam et al. |
| 17145 | BERT Loses Patience: Fast and Robust Inference... | Wangchunshu Zhou et al. |
| 18393 | A General Method for Robust Learning from Batches | Ayush Jain et al. |
| 18800 | Over-parameterized Adversarial Training: An An... | Yi Zhang et al. |
| 17964 | Reducing Adversarially Robust Learning to Non-... | Omar Montasser et al. |
| 18072 | Robust Federated Learning: The Case of Affine ... | Amirhossein Reisizadeh et al. |
| 18740 | Reliable Graph Neural Networks via Robust Aggr... | Simon Geisler et al. |
| 16819 | Towards More Practical Adversarial Attacks on ... | Jiaqi Ma et al. |
| 18236 | Boundary thickness and robustness in learning ... | Yaoqing Yang et al. |
| 17897 | Distributionally Robust Local Non-parametric C... | Viet Anh Nguyen et al. |
| 16965 | On the Trade-off between Adversarial and Backd... | Cheng-Hsin Weng et al. |
| 18190 | MetaPoison: Practical General-purpose Clean-la... | W. Ronny Huang et al. |
| 18869 | Certifiably Adversarially Robust Detection of ... | Julian Bitterwolf et al. |
| 18413 | Adversarially-learned Inference via an Ensembl... | Adarsh K Jeewajee et al. |
| 18283 | Perturbing Across the Feature Hierarchy to Imp... | Nathan Inkawhich et al. |