# OpenReview forum: "Adversarial for Good? How the Adversarial ML Community's Values Impede Socially Beneficial Uses of Attacks"
_ICML.cc/2021/Workshop/AML — ICML 2021 Workshop AML Poster_

### Official Review · Reviewer_8PHQ · 2021-06-19

**Rating:** Accept
**Confidence:** 4

**Review:**

This is an interesting paper studying the border impact of NeurIPS 2020 papers on adversarial machine learning. This paper examines the opinions of adversarial ML researchers towards adversarial ML for good, which is highly correlated with the topic of this workshop. The important finding is that most researchers do not take adversarial ML for good into consideration and mainly consider the security/robustness issues. A potential impact of this paper and also this workshop would be arousing the attention of more researchers in this field to realizing the positive aspects of adversarial ML beyond robustness.

---

### Decision · Program_Chairs · 2021-06-21

**Decision:**

Accept (Poster)

**Comment:**

This paper studied the border impact of NeurIPS 2020 papers on adversarial ML. The findings are interesting and may be insightful for future research.